# ASGR1 and Its Enigmatic Relative, CLEC10A

**DOI:** 10.3390/ijms21144818

**Published:** 2020-07-08

**Authors:** J. Kenneth Hoober

**Affiliations:** Susavion Biosciences, Inc., Tempe, AZ 85281, USA; jkhoober@susavion.com; Tel.: +1-480-921-3795

**Keywords:** lectin, asialoglycoprotein receptor-1, ASGR1, CLEC10A, macrophage, dendritic cell, T cell, tolerance, calcium, IL-10, IL-12

## Abstract

The large family of C-type lectin (CLEC) receptors comprises carbohydrate-binding proteins that require Ca^2+^ to bind a ligand. The prototypic receptor is the asialoglycoprotein receptor-1 (ASGR1, CLEC4H1) that is expressed primarily by hepatocytes. The early work on ASGR1, which is highly specific for N-acetylgalactosamine (GalNAc), established the foundation for understanding the overall function of CLEC receptors. Cells of the immune system generally express more than one CLEC receptor that serve diverse functions such as pathogen-recognition, initiation of cellular signaling, cellular adhesion, glycoprotein turnover, inflammation and immune responses. The receptor CLEC10A (C-type lectin domain family 10 member A, CD301; also called the macrophage galactose-type lectin, MGL) contains a carbohydrate-recognition domain (CRD) that is homologous to the CRD of ASGR1, and thus, is also specific for GalNAc. CLEC10A is most highly expressed on immature DCs, monocyte-derived DCs, and alternatively activated macrophages (subtype M2a) as well as oocytes and progenitor cells at several stages of embryonic development. This receptor is involved in initiation of T_H_1, T_H_2, and T_H_17 immune responses and induction of tolerance in naïve T cells. Ligand-mediated endocytosis of CLEC receptors initiates a Ca^2+^ signal that interestingly has different outcomes depending on ligand properties, concentration, and frequency of administration. This review summarizes studies that have been carried out on these receptors.

## 1. Introduction

ASGR1 and CLEC10A are members of a large group of proteins called ‘lectins’ that non-enzymatically bind carbohydrate structures. In plants these proteins usually occur in storage tissues such as seeds. Research over the past four decades identified a vast number of lectin receptors in mammalian tissues, which function as surveillance of the extensive glycome in normal tissues, recognition of environmental danger signals, cell-cell adhesion, ligand-specific endocytosis, glycoprotein turnover, and mediators of inflammation and immune responses. Five major families of lectin-type receptors have been cataloged: the I-type Siglec receptors are specific for glycan ligands that contain terminal sialic acid [1,2,3]; galectins bind ligands that contain β-galactoside moieties [4,5]; F-type lectins bind fucose [6]; and R-type lectins resemble the protein ricin [7]. By far the largest and most diverse family, indeed a superfamily, consists of C-type lectins that in most but not all cases require Ca^2+^ to bind the ligand within a conserved carbohydrate-recognition domain (CRD) [8,9,10]. Starting from the late 1980s, research on these lectins identified 17 sub-groups based on structure of the CRD and specificity of the sugar ligand [10,11,12].

Among the large family of Ca^2+^-dependent lectin receptors, two are of particular interest. The prototypic C-Type Lectin Domain Family 4 Member H1 (CLEC4H1, asialoglycoprotein receptor-1, ASGR1) and C-Type Lectin Domain Family 10 Member A (CLEC10A, macrophage galactose-type lectin, MGL, CD301) contain homologous CRDs that bind *N*-acetylgalactosamine (GalNAc) with much higher affinity than galactose (Gal). ASGR1 is expressed nearly exclusively by hepatic parenchymal cells but also by cells in the dermis of human skin [13] and at a much lower level in human peripheral blood monocytes [14]. Most of the early functional information on C-type receptors was developed with ASGR1, which functions to survey blood cells and plasma proteins for removal when sialic acid, the terminal sugar of attached glycans, is lost. CLEC10A is expressed at several critical stages of human development, with the highest expression in oocytes [15]. The totipotent 8-cell stage of embryogenesis expresses CLEC10A at a level 70-fold higher than later embryonic stem cells [16], and a three-fold increase in expression occurs upon differentiation of mouse multipotent stem cells to myeloid committed progenitor cells [17]. CLEC10A is also expressed on immature dendritic cells (DCs) [15,18], monocyte-derived DCs (mo-DCs) [19,20], alternatively activated M2a macrophages [21,22], and at low levels in human intermediate (CD14^++^CD16^+^) monocytes [23]. GalNAcα1-3Galβ is the most potent ligand for human circulating primary DCs [24]. CLEC10A mediates various immune responses dependent on the microenvironment and the nature of the ligand [19,25,26].

## 2. ASGR1

Gilbert Ashwell and his colleagues discovered ASGR1 during studies of the remarkably active uptake of serum glycoproteins by liver cells [27,28,29,30,31]. While most serum proteins contain glycan chains with terminal sialic acid, loss of this terminal sugar over time and exposure of penultimate Gal residues leads to rapid internalization by liver cells and degradation of these asialoglycoproteins. Orosomucoid (α_1_-acid glycoprotein) was the most actively internalized protein after removal of terminal sialic acid residues by treatment with neuraminidase [28]. Orosomucoid contains 183 amino acids (~20 kDa) but its mass is doubled by five N-linked glycan structures [32,33]. Ashwell’s team revealed the major characteristics of ASGR1, which include (i) a preference for terminal α-D-GalNAc (for example, a 60-fold greater affinity for GalNAc over Gal is expressed by the rat ASGR1) [34,35], (ii) increased affinity as a function of multivalency of the ligand, (iii) the number of receptors per cell, (iv) the overall rate of endocytosis (cellular uptake of the protein), and (v) receptor recycling.

ASGR contains two subunits, of which the predominant subunit, ASGR1 (CLEC4H1), contains 291 amino acids, whereas the less abundant ASGR2 (CLEC4H2) contains 311 amino acids, but the two are surprisingly different in sequence and in binding specificity. ASGR2 has only a two-fold selectivity for GalNAc over Gal, which is evident from the sequence of the CRD, and is easily dissociated from ASGR1 oligomers [35]. Various oligomeric structures were described, from the functional receptor occurring as homo-oligomers [36], a 2:2 heterotetramer [37], to a 3:1 heterotetramer [35]. Hardy et al. [38] observed two independent classes of binding sites for asialo-orosomucoid, described by K_D_ values of 0.44 nM and 9.7 nM. A synthetic ligand that contained six nonreducing Gal residues provided K_D_ values of 0.63 nM and 25.3 nM for the two sites, which, respectively, accounted for 817,000 and 1.23 × 10^6^ sites per rabbit hepatocyte. These studies were performed at 2 °C to prevent internalization of the receptor. Steer and Ashwell [31] had estimated 1.2 × 10^6^ total binding sites for asialo-orosomucoid per rat hepatocyte, whereas 200,000 to 500,000 receptors were detected on the surface of these cells [39,40]. The number of receptors on the surface at any one time depends on the extent of endocytosis and recycling [40,41].

The endocytic process was studied extensively by Schwartz et al. with asialo-orosomucoid [39,40,41,42,43]. Although Gal itself is a low affinity ligand for the receptor, multivalency—five terminal Gal residues per protein molecule or more, depending upon the extent of branching of the glycan—enhances avidity by orders of magnitude [43]. Even without a ligand, the receptor is internalized with a half-time of 5 to 6 min. In the presence of ligand, internalization is more rapid, with a t_½_ of 2.5 to 3 min, and the receptor returns to the cell surface in 5 to 7 min [44]. This remarkably active process of continuous recycling involves internalization with clathrin-coated pits [45], dissociation of ligand and receptor in the endosome concomitant with vesicle uncoating, and return of the receptor to the surface. The ligand is funneled through the vacuolar system to its fateful degradation in lysosomes [42,43]. Products of degradation of the asialo-orosomucoid appear in the medium about an hour after initial receptor binding [41]. Interestingly, there was a rapid loss of ASGR1 from the surface when cells were treated with weak bases such as chloroquine or primaquine, which inhibit acidification of the endosome and the return of the internalized receptor to the cell surface [44].

ASGR1 requires three bound Ca^2+^ ions to bind a ligand, and thus, dissociation of ligand in the endosome must be preceded by dissociation of the cation from the receptor. Whereas the initial insertion of Ca^2+^ occurs during folding of the protein within the endoplasmic reticulum, the continuous recycling of the receptor over several hours indicates that the CRD is recharged repeatedly with Ca^2+^ upon exposure to the extracellular fluid. This intuition is supported by the refolding of recombinant CRD of ASGR1 after expression in *Escherichia coli.* The inclusion body that resulted was solubilized in 8 M urea containing β-mercaptoethanol and then dialyzed against buffer containing 25 mM CaCl_2_ [46]. Although the three disulfide bonds that stabilize the CRD structure are apparently maintained during recycling, they form correctly from the completely unfolded structure after bacterial expression.

The extracellular concentration of Ca^2+^ is 1 to 2 mM, whereas the cytosolic concentration is less than 0.1 µM. During endocytosis, the Ca^2+^ concentration within the endosome drops rapidly [47], and as it is lowered below about 330 µM, the K_D_ of the two ‘high’ affinity Ca^2+^ binding sites in the CRD and with a reduced pH of 6.9, the ligand dissociates. This condition is achieved in about 3 min [47,48]. These events are dependent on acidification of the endosome by vacuolar H^+^-ATPase (V-ATPase) on the endosomal membrane [49,50]. Acidification of the endosome is a prerequisite to the reduction in the endosomal Ca^2+^ concentration, which suggests that an exchange mechanism is engaged to transport Ca^2+^ into the cytosol [51,52,53]. Although the cell has several mechanisms to control cytosolic Ca^2+^ concentrations near 0.1 µM, a transient elevation in the concentration induced by ligand-mediated recycling of the receptor initiates Ca^2+^-dependent signaling pathways within the cell.

## 3. Targeting Liver Cells via ASGR1

Extensive studies have been performed to find ligand conjugates that will provide specific transfer of drugs into hepatocytes through ASGR1. A basic structure of a tri-GalNAc ligand was initially synthesized by Lee and Lee [54]. Other teams further investigated multivalency and modifications of the ligand, and even polymers of GalNAc [55], to deliver various ‘cargoes’ into the liver [56,57,58]. Khorev et al. [59] designed trivalent structures with polypropylene or peptide spacers to extend the GalNAc residues into a space that would fit a trimeric receptor complex. The IC_50_ of binding of these ligands to the receptor follows the increasing avidity with valency, with mono-, di-, tri-, and tetravalent structures found to have K_D_ values of approximately 1 × 10^−3^, 1 × 10^−6^, 5 × 10^−9^, and 1 × 10^−9^ M, respectively. Cargoes attached to these structures for delivery into liver cells include radiolabeled human serum albumin or asialofetuin to measure liver function [60,61]. The structure-activity relationship between the optimal number of GalNAc residues and the linkage to the cargo have been extensively evaluated [62]. Recent advances involved GalNAc to deliver siRNAs, anti-microRNAs and antisense oligonucleotides for gene silencing in hepatocytes [62,63,64,65,66]. Initial clinical phase I trials found conjugates of a tri-GalNAc construct with antisense oligonucleotides to be highly potent and with a high margin of safety [67,68]. Ligand structures based on peptides, Gal or pullulan (a polysaccharide consisting of maltotriose units) were developed to deliver chemotherapeutic drugs such as doxorubicin and paclitaxel to treat hepatic cancers [69,70,71]. Complex structures in which GalNAc is attached to the tyrosine hydroxyl oxygen of proteins or peptides also express a K_D_ in the nanomolar range [72]. Studies of the size of the ligand indicated that particles with a diameter greater than 70 nm could not be processed by ASGR1 [73].

## 4. CLEC10A

In the late 1980s a lectin similar to the hepatic ASGR was found on mouse macrophages and designated the macrophage lectin specific for Gal and GalNAc (macrophage asialoglycoprotein-binding protein or M-ASGP-BP) [74]. This receptor was also called the mouse macrophage Gal and GalNAc-specific lectin (MMGL) [75]. Irimura’s team extended these studies, which were described in an extensive series of reports, and found that the mouse contains two related lectins, one specific for Gal (macrophage Gal-type lectin, MGL1) that is expressed predominantly by macrophages, while the other specific for GalNAc (MGL2) is expressed primarily by DCs [76,77]. A human protein (hMGL) was found that is homologous to the mouse MGL2 [78,79]. A powerful demonstration of the differential binding specificities of MGL1, MGL2, and hMGL was provided by Artigas et al. [80], who showed that hMGL effectively bound a glycopeptide containing one or two GalNAc adducts (e.g., the Tn antigen, *D*-GalNAcα_1_-*O*Ser/Thr) but not when Gal was added to GalNAc to form Gal-GalNAc-*O*Ser/Thr (the Thomsen-Friedenreich or T antigen). The human receptor (hMGL or simply MGL) bound to the glycan array in a similar fashion as murine MGL2, although the latter also bound to the T antigen with low affinity.

A receptor homologous but not identical to ASGR1 was discovered that is expressed by mo-DCs and was designated DC-ASGPR [81]. This receptor is 55% identical to ASGR1 but 82% identical in the CRD surrounding the QPD (Gln-Pro-Asp) sequence (boxed in Figure 1) that is correlated with Gal/GalNAc specificity [82,83]. DC-ASGPR is completely homologous to CLEC10A isoform 1 (MGL). (The National Center for Biotechnology Information does not include DC-ASGPR; the reported short form of DC-ASGPR [81] is C-type lectin domain family 10 member A isoform 2). CLEC10A has three major isoforms that are generated by alternative splicing. Isoform 1 contains 316 amino acids, whereas isoforms 2 and 3 contain 292 and 256 amino acids, respectively. Higashi et al. [79] found a total of seven isoforms in immature DCs and all contained the endocytic motif YENF (boxed in Figure 1) in the cytosolic domain. The gene for CLEC10A is located on chromosome 17 (17p13.1) and resides in a sequence of CLEC10A—ASGR2 with ASGR1 nearby.

These receptors have a strong preference for α-GalNAc over β-GalNAc, with very low binding to Gal [80,84]. CLEC10A is the only C-type lectin in the human immune system that exclusively binds terminal GalNAc residues [18,84,85]. CLEC10A was also found to bind tumor-associated sialylated-GalNAc antigens, Neu5Acα2,6-Tn, and NeuGcα2,6-Tn with similar affinities [86]. As with ASGR1, CLEC10A actively undergoes endocytosis. Most commonly, this process is analyzed by the disappearance of a tagged anti-CLEC10A antibody from the surface of DCs. The t_½_ of internalization of a bound antibody is about 5 min at 37 °C with mo-DCs [81,87,88]. Both ASGR1 and CLEC10A are type II transmembrane proteins, with the N-terminus in the cytosol. CLEC10A contains an endocytosis motif YENF near the N-terminus whereas the motif in ASGR1 is YQDL (boxed in Figure 1). Phosphorylation of the tyrosine residue (Y_5_, see Figure 1) is required for internalization [88] and subsequent signaling [89].

Within the immune system, CLEC10A is expressed by immature DCs [79]; CD1c^+^ DCs, for which CLEC10A is a specific marker [87]; mo-DCs [20,25,81]; M2a (alternatively activated) macrophages [21,90]; intermediate monocytes [23]; and at very low levels in many other cell types including monocytes, eosinophils, and erythroblasts [91,92,93]. CD1c^+^ blood DCs were recently subdivided into ‘non-inflammatory’ CD1c^+^A (DC2) and ‘inflammatory’ CD1c^+^B (DC3) sub-types, both of which express CLEC10A [94,95,96]. Moreover, CLEC10A is expressed by DCs that differentiate from monocytes that are recruited to inflammatory environments such as rheumatoid arthritis or ovarian ascites [20,97]. Expression of CLEC10A by DCs in the skin is most likely to detect pathogens containing GalNAc or Gal [98,99] and leads to strong Th2 or Th17 immune responses. Migration of the endocytic, active immature DCs to lymph nodes for interaction with T cells and initiation of an immune response has been found to require several days, during which time the DCs mature and express the co-stimulatory surface factors and MHC class II complexes that are required to activate T cells [100,101,102]. However, maturation of DCs leads to down-regulation of expression of CLEC10A [87,103].

Widespread interest has been focused on CLEC10A as an immuno-active receptor on immature DCs and alternatively activated macrophages [18,22,92]. As a pathogen-recognition receptor, CLEC10A binds to GalNAc in the teichoic acid of *Staphylococcus aureus* cell wall [104] and the surface of parasites such as helminths [84,105,106]. A common antigen on carcinoma cells, the Tn antigen (D-GalNAcα_1_-*O*Ser/Thr) [78,79,107,108,109], binds to CLEC10A with relatively low affinity (K_D_ = 8–12 µM) [85,110]. However, more complex GalNAc-containing ligands interact with additional groups within the binding site, which consequently increase the affinity by several orders of magnitude [111]. This feature was clearly demonstrated by Marcelo et al. [112], who found that a change from His^286^ to Thr^286^ (His^259^ in the sequence of their recombinant MGL) in the sequence His^284^-Phe-His^286^ (HFH^286^ in Figure 1) changed the specificity and lowered the affinity of Tn-peptide ligands. In particular, the sequence His^284^-Phe-Thr^286^ still recognized the Tn antigen but the receptor had much reduced ability to bind longer structures such as the blood group A structure. Moreover, whereas sialyl-Tn was bound by CLEC10A [86], the His^286^ to Thr^286^ mutation still bound the Tn antigen but eliminated binding of the sialylated ligand [112]. A ‘second’ binding site was thereby inferred that includes His^286^, which forms H-bonds with the underlying peptide or sugars in the ligand in addition to GalNAc [112]. Thus, the CRD of CLEC10A interacts with the carbohydrate structure and also the peptide backbone. Hepatic ASGR1 contains the analogous His^257^-Phe-Thr^259^ sequence (Figure 1), which may prevent the liver from actively ingesting young, sialylated red blood cells. The sequence in ASGR2 is Glu^257^-Val-Gln^259^ and the absence of His seems to reduce the specificity for GalNAc [111].

The highly glycosylated, cell membrane protein tyrosine phosphatase, CD45, which bears the Tn structure at positions 137 and 140 in exon B of the sequence [113], was identified as an endogenous ligand of MGL (CLEC10A) [103]. CD45 is expressed as several isoforms, with the full-length protein (CD45RABC) containing exons A, B, and C in the extracellular domain [114]. Binding of CLEC10A to exon B-containing isoforms causes attenuation of T cell activity, apoptosis, and immunosuppression [103]. However, activation of T cells leads to expression of shorter isoforms of CD45 such as CD45RO and CD45RA that lack exon B [115,116].

High avidity ligands for CLEC10A have been designed by increasing the number of GalNAc residues on MUC-derived peptides [19,25,111], which provide K_D_ values in the nanomolar range. Tn glycosylation of the MUC6 protein strongly affected its immunogenicity by diminishing a T_H_1 response while promoting a T_H_17/IL-17 response [117]. Multivalent Tn structures have been synthesized, built on a tri-lysine core, that contain 4 to 12 terminal GalNAc residues and have K_D_ values of 50 to 100 nM for binding to MGL [98,118,119,120], or simply polymers of GalNAc that have K_D_ values in the nanomolar range [55]. Multivalent Tn glycopeptides induced strong stimulation of CD4^+^ T_H_2 responses, with secretion of IL-4, IL-5, IL-13, and IL-10 and high levels of antibodies specific for the tumor-associated Tn antigen [98,118]. These ligand constructs that include the Tn antigen take advantage of the natural sugar as the specific ligand for CLEC10A [118,119,120]. 

## 5. A Ligand Mimetic

A tetravalent peptide with arms that mimic GalNAc was designed as a ligand for CLEC10A. The peptide has a K_D_ of binding to the receptor in the low nanomolar range, with an optimal concentration in cell cultures of 10 nM [121]. The arms of the peptide have the sequence NQHTPR (Asn-Gln-His-Thr-Pro-Arg), which is linked to a tri-lysine core through the sequence GGGS (Gly-Gly-Gly-Ser). The 6-mer sequence (sv6D) is the C-terminal half of a 12-mer sequence (svL4) originally identified through a screen of a phage display library with the GalNAc-specific lectin from *Helix pomatia*. The 12- and 6-mer peptides effectively suppress ovarian ascites, which is an inflammatory environment that recruits monocytes that differentiate into DCs that express CLEC10A [20].

Although non-antigenic in the mouse, antibodies were raised against sv6D conjugated to keyhole limpet hemocyanin (KLH). This antibody preparation recognized GalNAc conjugated to polyacrylamide, which demonstrated the authenticity of the peptide as a mimetic of GalNAc [121]. The predicted characteristics of binding of the peptide to CLEC10A and ASGR1 are similar to those of GalNAc. In silico modeling suggested that the peptide is larger than a single GalNAc unit but binds to CLEC10A in the same site as the sugar (Figure 2A,B). The QPD (Gln-Pro-Asp^269^) sequence that correlates with specificity for GalNAc/Gal, in contrast to EPN (Glu-Pro-Asn) that determines mannose binding in the mannose receptor [82,83,122], lies at one end of the binding pocket in CLEC10A. The guanidino group of Arg is in close contact with the sidechain carboxyl of Asp^269^ in the model of the receptor (Figure 2B). His^286^ provides preference for GalNAc in an engineered mannose binding protein [82,111,122] and lies under the peptide bound to CLEC10A. His^274^ is in contact with the peptidic His, while Trp^271^ lies behind the His-Thr-Pro-Arg sequence of the peptide. Interestingly, the oxygen atoms of the peptide are oriented outward while the ‘back’ of the peptide, in contact with the receptor, is largely hydrophobic and shielded from the Ca^2+^ ion by the Trp^271^ and His^274^. The predicted binding energy, ΔG′ = −41.6 kJ/mol, is derived from peptide/protein interactions [123]. Although the model predicts that the peptide does not directly interact with the Ca^2+^ ions, binding occurs in a strictly Ca^2+^-dependent manner [121].

A demonstration of the specificity of binding of peptides to CLEC10A in mo-DCs is shown in Figure 2C. Biotinylated peptide ligands of CLEC10A, sv6D, and svL4 [121], were incubated with a lysate of cells and then retrieved with streptavidin-coated magnetic beads, which pulled out a single protein. The calculated molecular mass of unglycosylated CLEC10A is 35,446 Da. With the two identified N-linked glycans, the mass is expected to be about 41 kDa. These cells are known to express DC-SIGN [126,127] and the mannose receptor [128], but CLEC10A was the only protein detected by this technique. In particular, a protein with the mass of DC-SIGN (45,775 Da plus the single N-linked glycan, or about 48 kDa) was not detected. CLEC10A was not detected in lysates of THP-1 monocytes (data not shown), but this protein was detected at very low levels on the surface of unstimulated THP-1 cells by flow cytometry [91]. Analyses of expression by single-cell RNA sequencing indicated a low level in primary monocytes [20,94].

## 6. Roles for CLEC10A in Health and Disease

Vlismas et al. [16] observed that human embryos at the totipotent eight-cell stage expressed CLEC10A at least 70-fold higher than the levels in embryonic stem and pluripotent cells. Klimmeck et al. [17] found that differentiation of hematopoetic multipotent progenitor cells to myeloid committed progenitors in the mouse was accompanied by a several-fold increase in expression of MGL2 (CLEC10A). The function of the receptor at these early stages of development is not known. Interestingly, a large population of ‘undifferentiated’ cells occurs in the peritoneum of Balb/c mice, which essentially disappeared when the animals were injected with svL4, a peptide ligand mimetic of MGL2. Concomitant with loss of the undifferentiated cells were increases in populations of mature immune cells [121]. In contrast, the undifferentiated population was minimal in C57BL/6 mice but increased dramatically within 24 h after an injection of svL4. After a second injection of svL4, the loss of this population was accompanied by increases in mature immune cells. The undifferentiated population was Lineage negative and did not stain for markers of mature cells. It was concluded that a common myeloid progenitor population resides in the peritoneum of Balb/c mice that differentiates to mature immune cells upon subcutaneous injection of a ligand for CLEC10A, and that this population can be induced in C57BL/6 mice [121]. Therefore, it appears that CLEC10A mediates proliferation of progenitor cells at least at two steps in the developmental pathway of mature immune cells in the peritoneal cavity of mice. The consequence of treatment with the peptide is expansion of the innate immune system, which may thus serve a foundational immunological role. The question is the specific action of CLEC10A/MGL2 in these processes.

An initial consequence of introduction of a ligand of CLEC10A is stimulation of endocytosis, with a concomitant influx of Ca^2+^. In studies with fibroblasts, endocytosis of fluorescent dyes (non-specific ligands) contributed a significant amount of Ca^2+^ to the cellular interior at a rate as much as 2 µM min^−1^ [47]. Acidification of the endosome, driven by vacuolar H^+^-ATPase (V-ATPae) [49,50] is a prerequisite to the reduction in the endosomal Ca^2+^ concentration, which suggests an exchange mechanism is engaged to transport Ca^2+^ into the cytosol [51,52,53]. Umemoto et al. [129] discovered that an influx of Ca^2+^ stimulated mitochondrial ATP production, which led to cell division of hematopoietic stem cells in C57BL/6 mice. These cells were described as ‘stem cells’ (Lin^-^ Sca-1^−^), which may be analogous to the myeloid committed progenitor cells described by Klimmeck et al. [17].

Several studies found a correlation between expression of CLEC10A and positive or negative outcomes in disease. Dusoswa et al. [109] found that glioblastoma tumor cells express relatively high levels of the Tn antigen. Moreover, CLEC10A is expressed at higher levels by tumor-associated macrophages and microglial cells in the brain. When a murine glioma cell line was engineered to further increase the expression of Tn, growth of tumors from the implanted cell line was enhanced. The increased number of infiltrating macrophages that expressed MGL (CLEC10A) also expressed PD-L1, both of which are immunosuppressive factors and were considered the cause of enhanced tumor growth. Similarly, Kurze et al. [130] found increased expression of Tn antigen in breast cancer tissues in mice treated with 4-hydroxy-tamoxifen, which increased phagocytic activity of macrophages that expressed CLEC10A within tumor tissue. However, in this case, the increase in Tn was associated with improved survival attributed to removal of damaged and dead cells.

Interestingly, house mite-induced dermatitis in mice is exacerbated in MGL1 (CD301a)-deficient strains of mice [13]. Repair of a point mutation in the gene for MGL1 by CRISPR attenuated severity of the dermatitis. Reporter cells that expressed MGL1 responded to a glycoprotein extracted from the mites that contained Tn and T structures, which ameliorated secretion of inflammatory cytokines. Whereas dermatitis caused a dramatic thickening of the epidermis, treatment with this glycoprotein reduced dermatitis induced by lipopolysaccharide in wild-type mice and restored the epidermis to a more normal thickness. Kanemaru et al. [13] proposed that ASGR1 is the human ortholog that defends against dermatitis. In a similar condition, MGL1 was upregulated in the lungs of a murine model of pneumonic sepsis caused by infection with the gram-negative bacterium *Klebsiella pneumoniae.* Deficiency of MGL1 resulted in significantly greater mortality, which indicated that MGL1 is required for resolution of pulmonary inflammation. Thus, MGL1 appeared to play a protective function in this bacterial infection [131], although the mechanism by which the receptor performs this function is not known.

## 7. Activation of T Cells

Tolerance defines a state of T cells. Whether T cells become tolerant is determined by the signal(s) given by DCs. Because CLEC10A is expressed by DCs and is often described as a “tolerogenic” receptor [18,109,119,132,133], it is useful to briefly review the outlines of T cell activation.

After phagocytosis by a pathogen-recognition receptor, peptide products generated by digestion in the endolysosomal system within DCs are loaded onto major histocompatibility complex (MHC) class II complexes and transported on vesicles to the cell surface. A functional presentation of these digestion products to the T cell receptor (TCR) on naïve CD4^+^ T cells requires three signals in correct sequence [134]. Signal 1 is recognition by the TCR of the antigen (’danger signal’) [135] presented by the MHC class II complex. Consequently, the TCR on T cells that recognize the antigen is phosphorylated on its cytoplasmic ITAM (immunoreceptor tyrosine-based activation motif) sequence, which leads to an increase in intracellular Ca^2+^. Signal 2 is expression of co-stimulatory factors CD40/CD80/CD86 by DCs as they migrate to draining lymph nodes where they interact with CD28 on the T cell. Furthermore, signal 3 is an inflammatory stimulus via secretion by DCs of cytokines such as IL-12, the principal cytokine for a T_H_1 response, type I interferon (IFN-α/β), or IL-27 [136] that amplify T cell differentiation and expansion [137,138,139]. Secretion of the type I interferons is an expression of a danger signal emitted by stressed cells or cells with damaged DNA that provide the essential context for an immune response, without which T cell receive a tolerogenic signal from DCs [140,141].

Upon binding of an antigen (signal 1), the ITAM sequence in the cytoplasmic domain of the CD3 subunit of the TCR is phosphorylated by the Src protein-Tyr kinases Lck and Fyn, which leads to subsequent phosphorylation of ZAP70 [142]. ZAP70 activates phospholipase C-γ1 (PLC-γ1}, which hydrolyzes phosphatidylinositol 3-P (PIP3) to diacylglycerol (DAG) and inositol 1,4,5-triphosphate (IP3), both of which are required to activate T cells. PIP3 is generated by PIP3 kinase, which is activated by CD28 on T cells by interaction with its ligands CD80/CD86 on DCs (signal 2). IP3 binds to its receptor on the endoplasmic reticulum and causes release of Ca^2+^, which binds to calmodulin and activates the Ca^2+^-dependent Ser/Thr phosphatase, calcineurin, which in turn dephosphorylates NFAT [142,143,144]. Translocation of dephosphorylated NFAT into the nucleus allows binding of the transcription activation factor to specific promoter regions and induction of expression of activation genes. DAG activates protein-Tyr kinase C-θ (PKCθ), which phosphorylates MAP kinases, including JNK2, which phosphorylates the Jun/Fos subunits of AP-1. AP-1 binds to and stabilizes the association of NFAT within regulatory elements of activating genes. PLC-γ1 and the Ca^2+^ arm lead to activation of NFAT, whereas PCKθ activity is required for AP-1, which in combination with NFAT induces expression of activation genes [144]. AP-1 is the key link between chromatin opening and activation of naïve T cells [145]. Reduced levels of AP-1 lead to loss of expression of activation genes and sustained unresponsiveness of the cells to subsequent stimulation by promoting expression of inhibitory genes by NFAT alone [144,145,146,147]. Whereas antigen presentation to the TCR activates the Ca^2+^ arm of T cell activation, treatment with anti-PD-1 compensates for a weak or absent signal 2.

Tolerance is achieved by several mechanisms. Lack of co-stimulatory signals such as CD86 leads to the absence of active AP-1, which leads to anergy and deletion, or bona fide tolerance, as described by Steinman [148]. Heissmeyer et al. [142] showed that a sustained elevated Ca^2+^ signaling induces a state of unresponsiveness in T cells by calcineurin-mediated degradation of PLC-γ1 and PCKθ. Without AP-1, NFAT locks in an inhibitory transcriptional pattern of genes [142,144,145,146,147], which leads to anergy. A lengthier process, described as ‘exhaustion,’ is associated with loss of effector T cell function and altered transcriptional patterns [149]. Whereas anergy is achieved within a few days, development of exhaustion requires weeks. The exhausted state can be overcome by checkpoint blockade [150].

An important program within the immune system’s repertoire is an actively regulated, antigen-dependent state of immunosuppression or an anti-inflammatory response that evolved to keep the immune system from over-reacting [141,151,152,153]. Upon ligation of CLEC10A on DCs, a signal transduction pathway is activated that leads to phosphorylation of CREB and expression of the *IL-10* gene [89]. IL-10 binds to the receptor IL-10R on T cells, which is associated with Janus kinase 1 (JAK1) and tyrosine kinase 2 (TYK2). Phosphorylation of these kinases leads to phosphorylation of tyrosine residues in the cytosolic domain of IL-10Rα, to which the transcriptional factor STAT3 is recruited. Activation of STAT3 by phosphorylation allows its translocation to the nucleus and initiation of expression of a set of genes that provide an anti-inflammatory response [154]. Interestingly, treatment of mo-DCs with a multivalent GalNAc-containing dendrimer significantly reduced expression of several genes encoding enzymes in energy metabolism and the rate of glycolysis [120]. This effect may also moderate the strength of the signals provided to T cells by DCs.

## 8. The Ca^2+^ Connection

Ca^2+^ is a critical regulatory messenger for many cellular activities [154,155]. Constitutive recycling of endocytic receptors such as ASGR1 [41], the heightened rate with a bound ligand, and transport of Ca^2+^ from the endosome to the cytosol [47,48,49,50,51], leads to an elevated cytosolic concentration [47]. Moreover, IgG-mediated phagocytosis through the receptor FcγRIIIb is accompanied by an increase in intracellular Ca^2+^ [156,157]. The maintenance of low intracellular concentrations of Ca^2+^ is determined by transport channels on the plasma membrane, the endoplasmic reticulum, mitochondria, and lysosomes [93,158,159,160]. Given that cellular responses may differ as a function of the Ca^2+^ concentration, we searched for clues that suggest how CLEC10A and other C-type receptors either induce tolerogenic or activation signals. The evidence suggests that high level engagement of the receptor (high occupancy) induces a high rate of Ca^2+^ influx and stimulation of a signal transduction pathway that leads to tolerance, whereas minimal occupancy (low concentrations of ligand) leads to activation. Full engagement of CLEC10A likely occurs when an antibody against the receptor is introduced, which leads to secretion of IL-10 [18,25,89,132,161]. A multivalent glycopeptide that contained 12 Tn (GalNAc) termini, added at a concentration of 5 µM, also induced secretion of IL-10 and TNF-α [98]. Other ligands of MGL (CLEC10A) often used to study DC function are the glycoproteins MUC1 and MUC6 or fragments of these proteins that have been modified to contain additional Tn antigens that bind the CRD. When added at a concentration of about 1 µM, Napoletano et al. [25] concluded that MGL (CLEC10A) engagement with the peptide containing 9 Tn adducts promoted DC activation. The responses were strongly dependent upon the type of ligand provided, with the Tn-MUC glycopeptide having advantages over the anti-MGL antibody for DC activation [26]. These studies show that a range of pathways, from activation to tolerance of DCs, can be induced with the appropriate conditions. Primary among these conditions is the ligand and the frequency and concentration of administration, i.e., the extent of engagement of the receptor [25,26] and the resulting Ca^2+^ response.

IL-10 is the main tolerogenic cytokine that keeps the immune system under control [153,154,161,162]. Increased secretion of IL-10 but decreased secretion of IFN-γ and IL-12 is indicative of a switch to an immunosuppressive (tolerogenic) profile. Within the reciprocal relationship between the anti-inflammatory IL-10 and pro-inflammatory IL-12 [163], extensive evidence has demonstrated that IL-10 is produced in response to an increase in intracellular Ca^2+^ [164,165,166,167,168]. In contrast, low concentrations of intracellular Ca^2+^ result in production of IL-12 [169]. Ligand-induced endocytosis of CLECs transfers Ca^2+^ bound to the receptor and in the extracellular fluid into the cell, where mechanisms described earlier transport Ca^2+^ from the endosome into the cytosol. Phosphorylation of tyrosine in the motif (YENF) in the cytoplasmic domain of CLEC10A [88], which is described as an endocytic motif as well as a hemITAM [170], initiates activation of a Src/Syk signal transduction pathway that includes PLCγ2, ERK1/2, p90RSK, and CREB and leads to expression of IL-10 [89]. A similar response was found in macrophages in which a Dectin-1/calmodulin-mediated activation of protein kinase II and Pyk2 led to phosphorylation of CREB and production of IL-10 [171]. Phosphorylation of the transcriptional factor, CREB, by calmodulin-dependent protein kinases mediates the response to Ca^2+^ and production of IL-10 [172,173].

This question can be further addressed with small ligands with which relative occupancy of the receptor can be manipulated. Eggink et al. [121] designed small peptide mimetics of GalNAc that bind to CLEC10A with high avidity (see Figure 2). Injection of the mimetic svL4 into glioma tumor-bearing mice at 1 nmole/g had little effect on reduction of tumor growth [174]. However, tumor size was an order of magnitude less in animals given 0.1 nmole/g doses for two weeks as compared with the untreated control (unpublished results). Tetravalent, short peptide mimetics of GalNAc, induced an initial (4 h after injection) T_H_1 type cytokine response in female Balb/c mice bearing an implanted breast tumor, in which the levels of IL-12 and IFN-γ were increased to a much greater extent than IL-10 [121]. The peptide dramatically extended survival of mice with an ovarian cancer cell line implanted in the peritoneal cavity, and doses of 0.1 nmole/g were more effective than 1 nmole/g [121]. In vitro experiments with human peripheral blood myeloid cells (PBMCs) showed that incubation with 10 nM sv6D caused a rapid (within 10 min) and extensive (up to 80%) dephosphorylation of a large number of signal transduction intermediates, followed by a rapid re-phosphorylation (unpublished results). This effect has the appearance of ‘resetting’ the signal transduction kinase intermediates to deliver an activation response. In contrast, 100 nM peptide did not cause a decrease in phosphorylation and only a slight rise in the phosphorylation state of several intermediates after 15 min. Activation by mo-DCs of secretion of IFN-γ by T cells was optimal at a concentration of 10 nM [121]. As shown in Figure 3, effects of peptides svL4 and sv6D on several processes in vivo, including proliferation of peritoneal cells, inhibition of ascites accumulation in a model of ovarian cancer, and reduction in the growth of glioma tumors, followed a bell-shaped curve. With the mouse, in vivo doses between 0.1 and 1.0 nmole/g seemed to correspond to the in vitro range of 10 to 100 nM.

Mayes et al. [175] presented a critical analysis of the potential dual activities of agonist immunotherapeutic antibodies. In contrast to antagonist drugs whose effects rise to a maximum at complete receptor occupancy with a sigmoidal dose-response curve, agonist drugs often induce maximal effects near 50% receptor engagement, with a reduction in effect at higher concentrations. Activities of the receptors are also strongly affected by the extent of clustering induced by the ligand. Westerfield and Barrera [176] proposed a mechanism for receptor activation based on ligand-induced clustering that determines efficiency and sensitivity. At high receptor occupancy, the receptor may aggregate to a greater extent, which would exclude CD45, the ubiquitous protein tyrosine phosphatase on immune cells, and allow the phosphorylated receptors to prolong the initiation signal. Clustering (dimerization) of the platelet receptor CLEC2 was required for phosphorylation by Syk/Src kinases of the hemITAM sequence (YITL) within the cytoplasmic domain and activation of downstream signaling, which involves PLCγ2 [177], a pathway similar to that for CLEC10A [89]. Ligand binding induced an increase in cytosolic Ca^2+^ that reached a maximal level about 90 sec after adding ligand, which was dependent upon the fluidity/stability of the membrane [177]. CLEC10A occurs on cell membranes as a trimer [178] and may form larger clusters with multivalent ligands. A bell-shaped pharmacodynamic curve for effects in the mouse as shown in Figure 3 has also been found with other Ca^2+^-mediated processes [179,180] and may be a common feature of the effects of C-type lectin receptors.

## 9. Conclusions

The cell biology of lectin-like receptors was established by the extensive early research with ASGR1, which set the stage for the work on C-type lectin receptors that followed. The role of C-type lectin receptors in the responses of immune cells to pathogens has been a critical motivator for research on these proteins. Although CLEC10A has been a research interest for many years, and a large amount of data has been obtained, the apparent pleiotropic effects induced by different environments and various ligands leave the action of this receptor as a conundrum. To achieve a deeper understanding of the activities of these receptors, it will be important to determine the extent to which activation of specific signal transduction pathways by C-type lectin receptors are dependent upon the cytosolic concentration of Ca^2+^ and the kinetics of transients of the cation [93]. There is a large body of evidence on the regulation of cellular events by Ca^2+^ on which these approaches can be based.

## Figures and Tables

**Figure 1 ijms-21-04818-f001:**
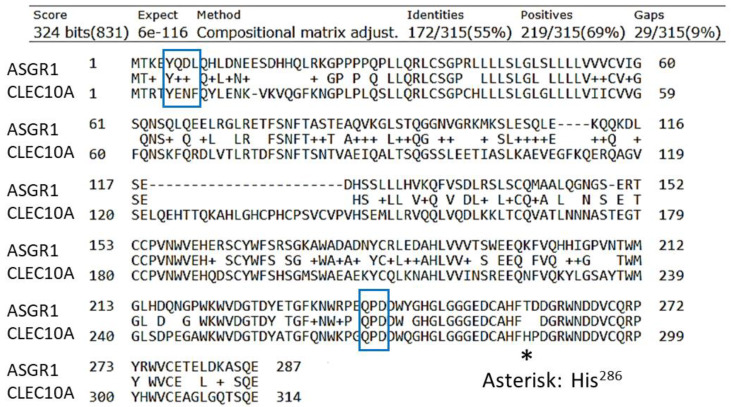
Comparison of the sequences of CLEC10A (isoform 1) and ASGR1. CLEC10A has two additional amino acids at the C-terminus, SH. ASGR1 has four additional amino acids, PPLL, at the C-terminus. The endocytosis motif (positions 5 to 8) and the QPD sequence that correlates with specificity for Gal/GalNAc are boxed.

**Figure 2 ijms-21-04818-f002:**
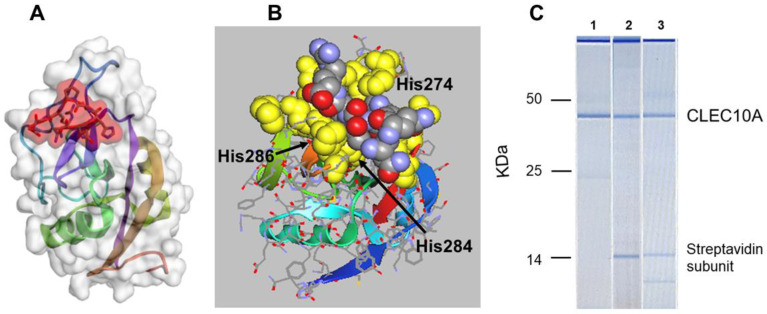
(**A**) In silico docking of an arm of sv6D (NQHTPR) to ASGR1 (accession number 1DV8) with CABS-dock (RMSD = 0.7611 Å) [124]. The peptide is enclosed in red shading. (**B**) The structure of CLEC10A was generated with SWISS-MODEL Deep View from the structure of ASGR1 [125]. Docking was modeled with CABS-dock (RMSD = 1.421 Å) and downloaded into ArgusLab. The position of sv6D in the binding pocket is shown after additional molecular dynamics. The peptide is colored (carbon, grey; nitrogen, blue; oxygen, red) while the binding site is yellow. The QPD sequence (Gln-Pro-Asp) is at the upper left of the binding site. The positions of His^274^, His^284^, and His^286^ of the binding site are indicated (see Figure 1). (**C**) A lysate of human mo-DCs was incubated with (1) mouse anti-human CLEC10A, which was recovered with magnetic beads coated with Protein A; (2) biotinylated sv6D; or (3) biotinylated svL4, which were recovered with magnetic beads coated with streptavidin. Proteins were eluted from the beads and subjected to electrophoresis with a BioAnalyzer (Agilent) in the presence of dithiothreitol. Molecular mass markers are indicated for IgG heavy chain (50 kDa), IgG light chain (25 kDa), and a streptavidin C1 subunit (13.6 kDa). The top band is an instrument marker.

**Figure 3 ijms-21-04818-f003:**
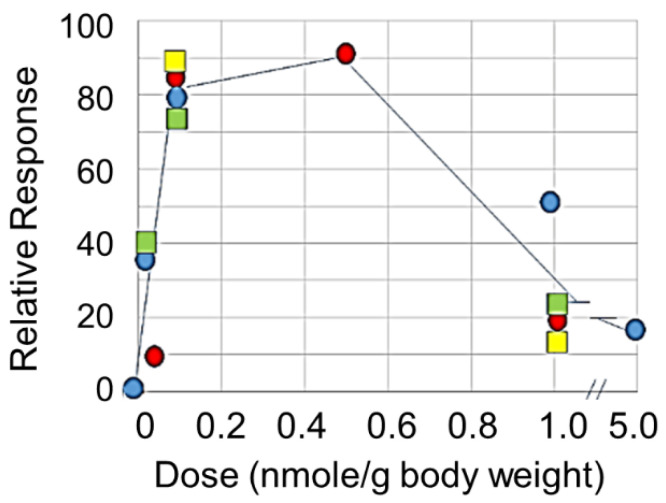
Effect of dose on response in C57BL/6 mice. The peptides were injected subcutaneously and measured endpoints included proliferation of progenitor peritoneal cells in healthy mice (red circles), inhibition of growth of glioma tumors with cells implanted in the brain (blue circles), and survival of mice with implanted ID8 ovarian cancer cells treated with svL4 (green squares) or sv6D (yellow squares).

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
