# Peer review of "ASGR1 and Its Enigmatic Relative, CLEC10A"

_ijms, 2020, doi:10.3390/ijms21144818_

Round 1

Reviewer 1 Report

In his review entitled "ASGR1 and its enigmatic relative, CLEC10A", Hoober reviews the current knowledge about two important C-type lectin receptors, the asialoglycoprotein receptor-1 (ASGR1), expressed by hepatocytes, and CLEC10A, mainly expressed by innate immune cells. The review is well written, excellently summarizes the state-of-the-art and also covers recent studies. The review is highly relevant and of interest for a wide readership. I recommend acceptance of the review article, provided that some minor points are considered (see below).

Specific points:

1.) The Introduction section needs revision. For instance, the author states that four major families of lectin-type receptors have been characterized (ll. 31 ff.). However, to my knowledge, there are more superfamilies, e.g. the F-type lectins etc. Generally, I do not see a reason why the author aims at briefly covering lectin families other than C-type lectin receptors at all (in less than one sentence each). Thus, I suggest to focus on C-type lectins exclusively in the Introduction section. In my opinion, it would even make sense to introduce the C-type lectin receptor superfamily more in-depth in this section.

2.) For the classification of the C-type lectins into sub-groups (ll. 34 ff.), the relevant first publications should be cited.

3.) Correction of some typos, e.g. "SIGLECs" (l. 37), "orosomucoid" (l. 81), "orosomucoid" (l. 89)... is needed (spell check).

4.) The section 7 "Activation of T cells" needs to be markedly shortened and streamlined. The author explains in detail the process of antigen presentation, T cell activation and T cell receptor signalling without a direct relation to CLEC10A (l. 345-388). In my opinion, major parts of this section can simply be removed or, at least, shortened. This also applies to Figure 3 that does not have a direct connection to the scope of the review.

5.) A comprehensive and continuative Conclusions and Perspectives section should be included in the review. Currently, the review ends quite abruptly. The only Conclusion is in lines 489-91, stating that "...there may be a way forward" (l. 4919). This section should be extended and should present connecting factors for further research.

Author Response

Response to Reviewer One:

I thank you for your kind words and careful review.

  1. My intention in the first paragraph of the Introduction was to briefly provide background context of the field of lectin-type receptors to indicate the relationship of C-type to other lectin receptors. I agree that too much was written about the other receptors and I shortened the list to a minimum to emphasize C-type lectin receptors (lines 31-38). 

  1. I included publications by Drickamer and colleagues (references 11 and 12) that describe early recognition of the C-type class of receptors and the beginnings of the cataloging into subgroups.

  1. I appreciate the typos pointed out, which motivated a thorough examination of the review for other misprints.

  1. The role of CLEC10A (MGL) in T cell tolerance has dominated discussions in the literature on this receptor. However, some data conflict with this conclusion, and a range of effects of dendritic cells on T cells has been observed. Thus I took a broader view to determine whether there are various effects of CLEC10A on DCs that would provide a more complex (complete) view of their interactions with T cells.  It is important to recall that CLEC10A is a pathogen recognition receptor that allows dendritic cells to activate T cells.  For this reason, I retained the text description on T cell activation (lines 355-393) but removed Fig. 3, the diagram on T cell activation, to shorten this section.   

  1. I appreciate the comment regarding the end of the review, which did end abruptly. I added several sentences as a Conclusion that reiterate the main theme of the review without being repetitive.  My major issue is the need for a greater focus on the role of Ca2+ in the activity of these receptors in immunology (lines 508-519).

Reviewer 2 Report

In his review manuscript dr. Hoober describes and compares two GalNac-specific C-type lectins, namely ASGR1 and CLEC10A. The review is extensive, well-balanced, as well as easy to read and comprehend, also for researchers outside of the C-type lectin field. I appreciate the additional information on T cell activation and tolerance, which will assist readers of IJMS. Also the Ca2+-connection and the hypothesis to explain both the stimulatory and tolerogenic properties of CLEC10A are certainly interesting.

I only have a few remarks and suggestions to further improve the manuscript:

  • The first paragraph on page 2 describes the expression pattern of CLEC10A. Please add here the current consensus that CLEC10A is an excellent marker to distinguish human CD1c+ dendritic cells in many tissues (Heger Front Immunol 2018).
  • The carbohydrate recognition profile of ASGR1, discussed on page 2, is that of rat Also the mentioned sensitivity of GalNAc over Gal refers to rat ASGR1. As many C-type lectins, including ASGR1, have different carbohydrate specificities in different species (for instance Park JBC 2004), this is somewhat misleading. Therefore, the author should make refer to the rat specificity in this section and also include the specificity of human ASGR1.
  • On page 4 the author describes the carbohydrate specificities of human MGL and mouse MGL2 and states “The human receptor (hMGL or simply MGL) 156 bound to the glycan array in strikingly similar fashion as murine MGL2.” This is not correct. The carbohydrate specificity of MGL2 is much broader compared to the human MGL. Whereas human MGL hardly bind galactose, MGL2 readily recognizes it and also binds for instance TF (Singh Mol Immunol 2009).
  • It is worth mentioning in the second paragraph of page 5 that in the mouse, MGL2 exclusively marks dendritic cells that are crucial for the establishment of type 2 immunity (Kumamoto Immunity 2013).
  • Targeting of antigens to CLEC10A+ dendritic cells leads to the differentiation of IL-10-producing suppressive CD4+ T cells (Li J Exp Med 2012), supporting the role of CLEC10A as a tolerogenic marker. I could not find this back in the manuscript, but is part of the story as one considers both the stimulatory as well as immune inhibitory properties of CLEC10A.
  • Lastly, I noticed a few typing errors:
    Line 26: “CLC10A” instead of “CLEC10A”
    Line 99:        “ASGPR1” instead of “ASGR1”
    Line 185:      “CD2” instead of “DC2”
    Line 187:     “type” instead of “types”

Author Response

Response to Reviewer Two:

I thank you for your kind words and insightful review.

  1. The listing in the Introduction of the cell types that express CLEC10A was a brief over-view. A more detailed description of the expression of CLEC10A is given on page 5 (section 4, lines 195-202), in which the reference to Heger et al. is included as number 87.   

  1. The reference to the specificity of the rat ASGR1 for binding to GalNAc and Gal (line 79) was intended as a general example. The biochemistry of this receptor has been extensively studied in mouse, rat and rabbit, but the simple binding data that I wanted to convey were provided most extensively with the receptor from rat. 

  1. To illustrate the specificities of MGL1, MGL2 and hMGL, I used the recent side-by-side array data by Artigas et al. (reference 80). I modified the sentences on page 4 (lines 161-166) to also indicate that MGL2 but not hMGL binds to the T antigen.

  1. Data that demonstrated that MGL2 is exclusively expressed by dendritic cells were included as Denda-Nagai et al. (2010) and Higashi et al. (2002) (references 77 and 79), which were definitive earlier publications from Irimura's laboratory on the expression of MGL2.

  1. The effect of inducing tolerance was a significant issue in this review. The publication by Li et al. (2012), indicated by the Reviewer, was included as number 161.  An important, more recent publication from this laboratory was included as reference 89, which described their work on the signal transduction pathway in dendritic cells to IL-10.

  1. I appreciate the typing errors found by the Reviewer, which were corrected.